# Point Shear Wave Elastography in Diagnosis and Follow-Up of Salivary Gland Affection after Head and Neck Cancer Treatment

**DOI:** 10.3390/jcm11216285

**Published:** 2022-10-25

**Authors:** Benedikt Hofauer, Andreas Roth, Clemens Heiser, Johannes Schukraft, Felix Johnson, Zhaojun Zhu, Andreas Knopf

**Affiliations:** 1Department of Otorhinolaryngology/Head and Neck Surgery, Klinikum Rechts der Isar, Technical University of Munich, 81675 Munich, Germany; 2Department of Otorhinolaryngology, University Medical Center Freiburg, 79106 Freiburg, Germany

**Keywords:** head and neck squamous cell carcinoma, point shear wave elastography, ultrasound, sicca

## Abstract

Therapies of head and neck squamous cell carcinoma (HNSCC), particularly irradiation and chemotherapy (C/RT), can affect salivary glands to some extent. Recent studies suggest that point shear wave elastography (pSWE) is well suited for the diagnosis and rating of homogenous damage to parenchymatous organs. The purpose of this study was to assess the value of this sonographic modality as a tool for the evaluation both of salivary gland affection after HNSCC therapy and the effect of a salivary replacement therapy based on liposomes. A total of 69 HNSCC patients were included in this study. All patients had finished cancer treatment and attended regular follow-up. pSWE values of ipsi- and contralateral parotid (PG) and submandibular glands (SMG) were obtained in a standardized manner and compared to those of a healthy control (HC) group. After a two months treatment with a liposomal saliva replacement therapy pSWE quantification was performed again. Ipsi- and contralateral salivary glands suffer under standard HNSCC tumor therapy. Here, the ipsilateral parotid gland (PG) is primarily affected. Therefore, a sequence of manifestation (surgery < surgery plus adjuvant C/RT < primary C/RT) is comprehensible, evaluated by pSWE measurements. The examination of all glands and statistical analysis of the values compared to controls resulted in an pSWE cutoff value for affected glandular tissue of >2 m/s. Using a liposomal saliva replacement therapy, pSWE values of the ipsilateral PG can be improved, although the level of HC could not be restored.

## 1. Introduction

Hyposalivation and consecutive xerostomia is one of the most significant long-term complications for more than 550,000 patients that are annually treated with head and neck cancer (HNC) [1,2] and for whom chemo-/radiation (C/RT) is one of the pillars of curative and palliative therapy [3]. The healthy tissue within the field of irradiation suffers from severe side effects of the necessary dosage to eradicate the tumor cells [4]. Recent work suggests that over 90% of patients with malignancies of the head and neck develop some degree of xerostomia after radiotherapy. Complaints such as oral discomfort, increased rates of dental caries and oral infection, taste disorders, as well as difficulty in speaking and swallowing are common and a source of major distress [5,6,7]. Even for complications as severe as osteoradionecrosis of the mandibula, the lack of salivary protection is a well-recognized contributing factor [8]. One reason for this complex of symptoms is C/RT-induced mucositis, which frequently occurs in patients with head and neck cancer [9,10,11]. Particularly chemotherapeutics, such as 5-fluorouracil and cisplatin, which are usually used in patients with squamous cell carcinoma of the head and neck, have proven mucotoxic effects [11]. The combination of radiation and chemotherapy may even produce additional damage to the oral mucosa and the salivary glands as well [12]. Both patient-reported and observer-rated xerostomia after C/RT are dose dependent, whereas the mean doses applied to parotid gland (PG), submandibular gland (SMG), and mucosa of the oral cavity (OC) were significant predictors. These results support efforts to spare salivary glands by C/RT using intensity-modulated radiotherapy [13]. It has been shown that not only C/RT in the head and neck region but also the specific primary tumor site (larynx/hypopharynx), surgical procedure, and treatment non-related independent factors, such as older age and advanced T-status, are correlated with the impaired function of the salivary glands and the swallowing apparatus [14]. Surgically treated, symptomatic patients have been included as an internal control to statistically rule out the contribution of lifestyle and the HNSCC itself to glandular alteration. This leaves the modus of treatment as matter of determination.

In order to assess the capability of pSWE to monitor the outcome of therapy, a local liposomal agent (LipoSaliva^®^, Optima Pharmaceutical GmbH, Wang, Germany) was applied to reduce cancer treatment-related side effects. A substantial improvement of clinical endpoints such as taste and smell through LipoSaliva^®^ has been shown before in a similar setting [9,15,16,17,18]. However, no objective testing is established yet that reliably quantifies the glandular destruction and monitors potential glandular restoration.

pSWE is a dynamic shear-wave speed measurement at a selected region of interest usually within an organ. It is a modern sonographic method which generates information on the stiffness of tissue. The shear wave is generated by application of an acoustic push pulses (a focused ultrasound beam, also called acoustic radiation force impulse, ARFI), which generates a standardized, localized displacement within the selected tissue region. This results in a lateral tissue displacement with a certain speed, which depends on the stiffness of the tissue examined. This lateral propagation is measured by the ultrasound machine using a Doppler-like ultrasound technique and is depicted as shear wave velocity (SWV) in meters per second (m/s) [19,20,21,22].

The aim of this study was to evaluate the role of pSWE as a non-invasive diagnostic tool that reliably identifies and evaluates glandular affection due to C/RT and use it to monitor parenchymal recovery of glandular tissue after local liposomal treatment.

## 2. Materials and Methods

This study was conducted at the Department of Otorhinolaryngology, Head and Neck Surgery, Klinikum rechts der Isar, Technical University of Munich. The study protocol was in accordance with the Declaration of Helsinki, revised in 2000, Edinburgh and approved by the local ethics board. Written informed consent was obtained from all participants. The study was approved by the local ethical committee (No. 4022/11).

During a period of nine months, 69 patients with squamous cell carcinoma of the head and neck (HNSCC) that suffered from xerostomia were consecutively included in the observation. Carcinoma in situ or other histological subtypes, patients who refused or abandoned therapy and patients with a history of multiple HNSCC treatment were excluded. Clinical data, e.g., age, sex, noxa status, TNM status, G and R status, and treatment, were collected following current clinical guidelines in all cases (UICC 7th edition). The patients had all finished tumor therapy and attended our clinic for regular, post-therapeutic follow-up.

Patients that received C/RT were treated according to clinical standards. For primary C/RT, patients were irradiated with 70 Gy to the primary tumor location and the draining lymph nodes received 70 Gy if affected and 50 Gy for prevention. In adjuvant C/RT, 50 Gy with boosts up to 64 Gy for the primary malignancy and 64 Gy for the draining neck region were applied. The salivary glands were spared by exclusion from the field of radiation if oncologically possible. Chemotherapy consisted of up to 6 cycles of 40 mg of Cisplatin.

The patients were divided into 3 different groups according to their treatment of HNSCC: group 1 just surgery, group 2 surgery and adjuvant C/RT and group 3 primary C/RT. Qualitative head and neck high-resolution B-mode ultrasound was performed for all patients during pre-therapeutic staging using a 9–14 MHz linear transducer (tissue harmonic imaging [THI]; Acuson S2000, Siemens Healthcare, Erlangen, Germany).

All patients underwent pSWE imaging of the PG and SMG on the same machine during a standard cancer follow-up visit. Measurements of pSWE for the entire study cohort were acquired at a depth of 1.5 cm in the center of the submandibular gland and the caudal pole of the parotid gland outside the jaw and large blood vessels, e.g., the temporal superficial artery. Measurements were obtained by experienced examiners (>3 years of experience, certified according to the German Society of Ultrasound in Medicine) following standardized protocol with moderate transducer pressure timed to the absence of swallowing maneuvers. pSWE was performed using ten individual measurements in each of the parotid and submandibular glands [19,23].

All patients received symptomatic local treatment with the liposomal mouth spray for a period of two months. Patients were instructed to use this spray three times per day with five bursts to the mouth. The salivary replacement was not blinded or placebo controlled, since the spray has a distinct taste of fat. Furthermore, it is not the pharmacological effect of liposomal treatment that is the focus of this work.

High-resolution B-mode ultrasound and pSWE was performed a second time in the same fashion as described above two months after local therapy. Equal measurements of the major salivary glands of 50 volunteers who were referred to our department for non-oncological and non-salivary-gland-related head and neck diseases were performed. Those patients did not use any saliva replacement substances and served as HC. All HC were screened for a current of past history of diseases affecting the salivary glands (oncological diseases, autoimmune disorders, other salivary gland disorders).

Receiver operating characteristic (ROC) analysis and Youden tests were conducted to define the cutoff of pathological major salivary gland induration in HNSCC patients following treatment. Data were analyzed via unpaired t tests and Chi-squared test, group analysis was performed via ANOVA and post hoc analysis was calculated via the Tukey b test using GraphPad Prism 6.

## 3. Results

A total of 69 patients were included in the study. The mean age of patients at cancer diagnosis was 57 years with standard deviation of ± 11 years and a majority of male over female patients of 2.84:1. The 50 HC were a mean of 58 ± 16 years of age and a sex distribution of 5.25:1 female to male. With 47 cases, the oropharynx was the most affected tumor localization. Six cases each were the hypopharynx and CUP syndrome of the neck, followed by five laryngeal and four nasopharyngeal carcinomas, respectively. One patient suffered from OC HNSCC. A third of the cases showed T2 status at diagnosis followed by 30% in T1 and 15% in T3. T4 status was diagnosed in 13% of the observed cases. Cervical lymph node metastases were seen in 67% of the patients, ranging from N1 to N2c status. No lymph node affection could be detected in 32% of the cases. Distant metastasis was seen in only one patient. Histological examination revealed G2 and G3 status in 96% of tumor samples, while 4% were graded G1 (Table 1).

The vast majority of patients (94%) who underwent surgery had a postoperative R0 status. Only in three cases (6%), a R1 status was histologically confirmed. Seventy-six percent of patients underwent surgery, and the other twenty-four percent underwent primary C/RT. Fifty-five percent of patients who underwent surgical approaches received adjuvant therapy, comprising 29 patients (42%) with adjuvant RT and 9 patients (13%) with adjuvant C/RT. A history of nicotine and alcohol abuse was reported by 59% of patients. Forty-five percent of patients demonstrated simultaneous alcohol and tobacco abuse (Table 1).

The median timespan between diagnosis and first ultrasonic examination was 26 ± 30 month. Stratification of different timespans from diagnosis to pSWE measurement (<25 month; 25–48 month; >48 month) did not show differences between the groups.

In comparison to the HC group, the patients showed a significant stiffening for the glands, especially of the ipsilateral PG and SMG after tumor therapy.

As depicted in Figure 1, the SWV in the ipsilateral PG was higher (2.31 ± 0.74 m/s) compared to the contralateral gland (2.2 ± 0.65 m/s), both being significantly higher compared to the PG of HC (1.87 ± 0.7 m/s, *p* < 0.001). The ipsilateral SMG had higher SWV (2.23 ± 0.67 m/s) compared to the contralateral gland (2.18 ± 0.54 m/s), both being significantly stiffer than the SMG of the HC group (1.81 ± 0.36 m/s, *p* < 0.05). To illustrate the glandular images taken to survey the pSWE values, see Figure 2.

Further analysis (ROC and Youden test) revealed an optimal diagnostic cut-off (sensitivity: 66%; specificity: 62%) of >2.0 m/s shear wave velocity for diagnostic discrimination (Figure 3). The PG ipsilateral to primary carcinoma demonstrated the highest SWE values after primary radio-chemotherapy (*p* < 0.001), followed by adjuvant radio-chemotherapy (Figure 4). The least glandular stiffness was observed in the ipsilateral PG of solely surgically treated patients. Intriguingly, this exact trend is reversed in observation of the contralateral (*p* < 0.0001). The affection of the ipsilateral SMG measured by pSWE follows the same pattern as the ipsilateral PG, but with lesser significance. Contralateral SMG stiffness shows no correlation to the form of cancer treatment administered (Figure 4).

All salivary glands observed have stiffened throughout cancer therapy in close vicinity resulting in a cut-off >2 m/s. After a two-month treatment period, SWV showed a significant decrease from 2.3 ± 0.66 m/s to 2.09 ± 0.55 m/s (*p* < 0.0001) but failed to restore the level of the HC group of 1.87 ± 0.7 m/s (*p* < 0.001). For all the other glands, we observed the same tendency but could not determine statistical significance (Figure 5).

## 4. Discussion

In general, any HNSCC therapy affects the salivary glands [1]. Subsequent xerostomia, dysphagia, and speech handicap dramatically decreases quality of life. Although a substantial proportion of HNSCC patients show glandular impairment, diagnostic tools that reliably identify and monitor post-therapeutic glandular impairment are missing. In this study, we show the objective quantification of the extent of damage caused by therapy through pSWE. As an easy-to-perform and readily available technique, pSWE offers a way of standardized categorization and correlation of the glandular stiffness and the patients’ symptoms within tumor therapy follow-up. Since a variety of tumor entities have been examined in this study, the distance of the tumor to the salivary glands also varies. This fact might account for the large variation in values as depicted in Figure 4. The sub summation of entities still makes sense because of the common side effect of xerostomia through affection of the salivary glands which is examined in this study—however, a sole inclusion, e.g., of patients with oropharyngeal carcinoma, would have resulted in a more homogenous patient cohort and this should be reconsidered in future projects. Ipsi- and contralateral PG and SMG have been examined and the results show that all four large salivary glands significantly increase in stiffness after all observed tumor therapies when compared to those of healthy individuals using pSWE. Most affected was the PG ipsilateral to the field of irradiation in patients who received any radio therapy compared to the contralateral counterpart. This seems consistent with the widely used irradiation technique of contralateral parotid-sparing intensity-modulated radio therapy (CLPS-IMRT). Notably, this does not explain the significant hardening of the contralateral PG after surgery we did observe (Figure 4). Some influence similar to an abscopal effect seen for tumors and their distant metastases after radiation therapy has been proposed for surgical interventions as well [24]. In our cases, though the salivary glands have not been directly operated on, the distant damage to the corresponding contralateral PG seems unlikely. Another possible explanation could be the post-surgical fibrous stiffening of the ipsilateral neck. This often leads to substantial overuse of the contralateral mandibular and muscles for chewing. This in turn could lead to an altered glandular structure above the retrained masseter muscle. However, this hypothesis needs further investigation.

Although serous glands are supposed to be more vulnerable than mucinous ones, our results show no significant difference between the SWV within ipsilateral PG and SMG. The values measured in our patient collective and the group of HC roughly reflects the ones obtained by colleagues in Romania in 2011. The group examined only the SMG of 27 HC and 18 irradiated patients and also found a significant difference in SWV. The mean of 1.82 m/s and 2.13 m/s in untreated and treated individuals, respectively, are consistent with our results concerning the SMG [25]. Therefore, SWV in SMGs of treated patients was above our postulated cutoff value of >2.0 m/s. The observed impairment of ipsilateral glands of treated patients is most significant for the ones that received primary C/RT. With decreasing dosage of radiation and systemic therapeutics such as in the adjuvant C/RT-treated patients, the stiffness of these glands was smaller. The least side effects on the ipsilateral glands could be observed in the patients with surgical treatment only. This dose dependent trend is adequately explained by the commonly known side effects of such therapy described in the pertinent literature. However, the results of our former clinical trial showed a significant decrease of taste and smell in patients who underwent surgery without adjuvant treatment [9]. This seems consistent with the results of the presented pSWE data, because we see a clear difference between the SWV of HC and patients that have undergone surgery only. We must assume that different pathophysiologic mechanisms such as a carcinogenic lifestyle or the immune response to HNSCC could be critical to explain this phenomenon. In addition, the opposite was observed in the contralateral PG, where the stiffness of the tissue decreases as the dosage of harmful therapeutic agents increases, which seems contradictory. As nerves and blood vessels aid in epithelial cell repair post-RT, the combined radiation damage to acini, ducts, nerves and blood vessels, and development of fibrosis obstructs, but also activates normal gland regeneration [3,26]. It might well be that the program of restoring the glandular structure and function by resident stem cell reservoirs has been generally activated in glandular tissue by treatment of the contralateral PG. However, the damage inflicted to the organ further away from the harmful treatment was beneath a certain threshold so that a damaging point of no return was not met and the glandular repair program could restore the tissue over time. Even alteration of lifestyle factors such as abuse of noxa, as many patients undertake after tumor diagnosis, could contribute to this finding.

Here, we statistically established an objective cutoff that quantifies the glandular destruction as side effect of loco regional tumor treatment and monitors potential glandular restoration. This can be easily obtained by sonography and provides a valuable tool for further examination of similar arrays of questions concerning the subject. However, it would be useful to prospectively pair the results with an objective measure of salivary production such as the unstimulated salivary flow (USSF) test and a subjective quantification of xerostomia by questionnaire.

Taken together this suggest that above the threshold of 2 m/s of SWV the ipsilateral PG and SMG show increasing side effects according to the dosage of irradiation or systemic chemotherapy. The side effects of surgery seem to be of less harm to the gland (Figure 4).

We further assessed the alteration of the glandular tissue using pSWE after topical therapy. The use of a liposomal saliva replacement resulted in a significant softening of the glands measured. In particular, the condition of the ipsilateral PG can be improved, although the pre therapeutic level of a healthy gland could not be restored. All the other glands improved as well, although not to the same degree of significance. Before, we were able to show the beneficial application of liposomal sprays in HNSCC patients who had already completed their tumor therapy within a clinical trial using clinical scores evaluating the olfactory, gustatory and a visual analog scale to quantify xerostomia [9].

The mechanistic reason for glandular improvement in patient-centered subjective and in objective measurements remains a matter of speculation. A possible mechanism might be the protection against harmful infectious bacteria and fungi of the oral flora, which might slowly damage the glands by causing retrograde subclinical chronic inflammatory stimuli. As a means of moisturizing and flushing the glandular ducts, saliva replacement could also establish a protective layer on the epithelial lining. To unfold such mechanisms in the future, more experimental work has to be done.

In conclusion, a substantial proportion of HNSCC patients demonstrate glandular impairment after successful cancer treatment. pSWE represents a non-invasive diagnostic tool that reliably identifies glandular impairment after HNSCC therapy and is able to monitor a response with regard to tissue stiffness after using local liposomal therapy. Further investigation should focus on correlation to salivary flow, symptoms and quality of life, as well as mechanisms of glandular damage and repair.

## Figures and Tables

**Figure 1 jcm-11-06285-f001:**
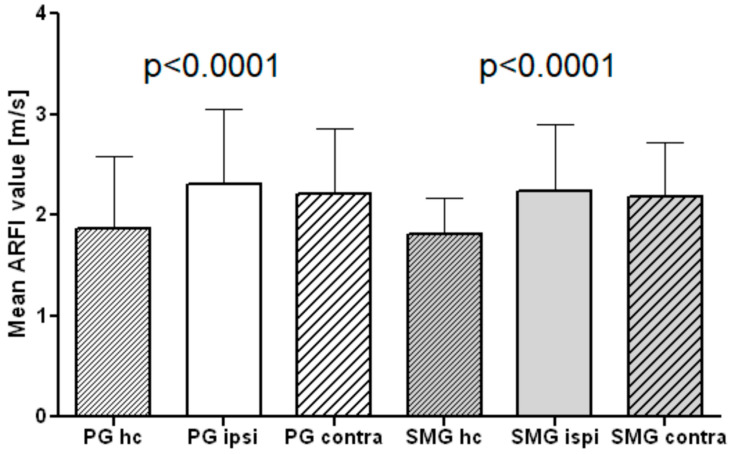
Point shear wave elastography (pSWE) imaging values of parotid glands (PG) and submandibular glands (SMG) ipsi- and contralateral of patients compared to 50 healthy controls (HC).

**Figure 2 jcm-11-06285-f002:**
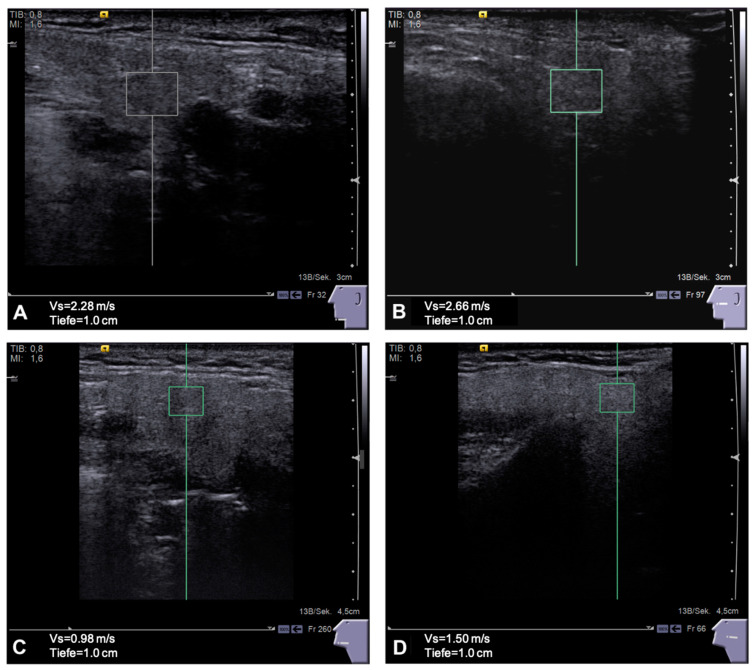
Point shear wave elastography (pSWE) images and corresponding shear wave velocities of a patients (**A**) GSM 2.28 m/s and (**B**) PG 2.66 m/s are shown. Glands are notably smaller and more fibrous after surgery and adjuvant C/RT. The respective healthy glands and values of a control person are shown in (**C**) GSM 0.98 m/s and (**D**) PG 1.50 m/s.

**Figure 3 jcm-11-06285-f003:**
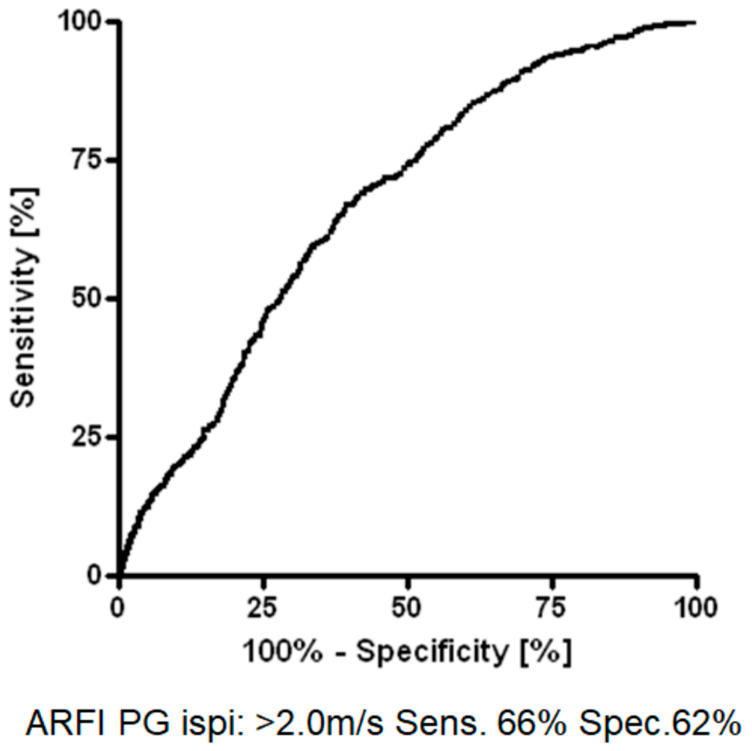
Receiver operating characteristic (ROC)-analysis and Youden test of shear wave velocity within the ipsilateral parotid glands (PG).

**Figure 4 jcm-11-06285-f004:**
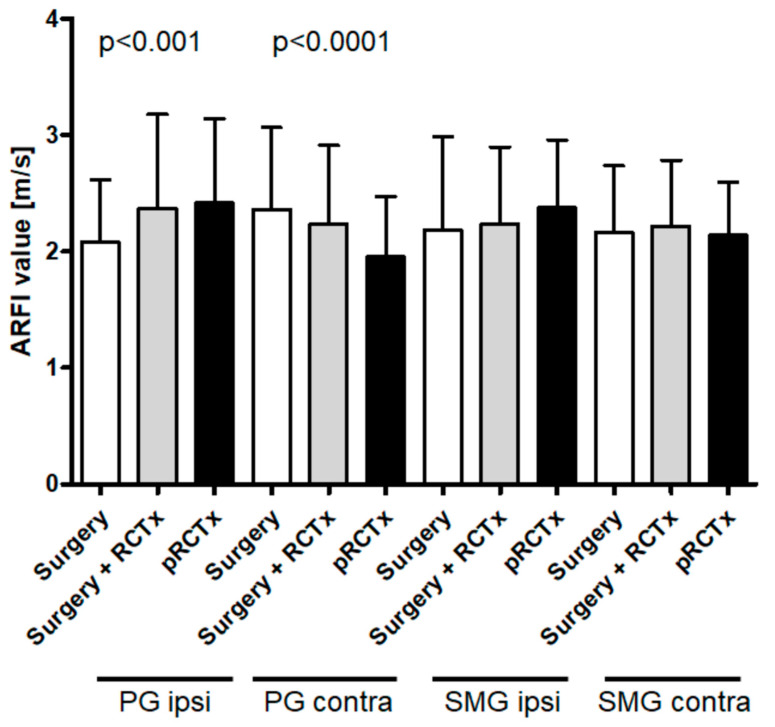
Stratification of the affection of ipsi- and contralateral salivary glands dependent on the therapeutic regime.

**Figure 5 jcm-11-06285-f005:**
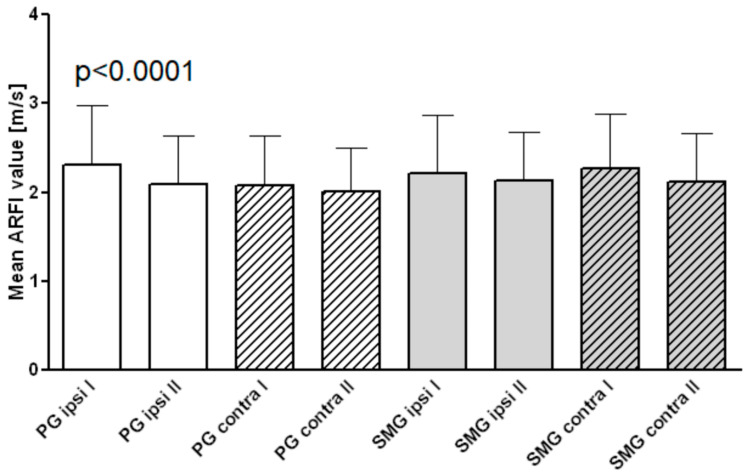
Point shear wave elastography (pSWE) imaging values of salivary gland tissue after tumor therapy before (I) and after (II) 2 months of topical liposomal therapy.

**Table 1 jcm-11-06285-t001:** Patient characteristics, tumor staging and grading and history of substance abuse. All data are expressed as median (mean ± standard deviation) or number (%).

N	69
**Age at tumor diagnosis (years)**	57 (57 ± 11)
**Sex**, Male/Female	51 (74)/18 (26)
**Localization, n (%)**	
Nasopharynx	4 (6)
Oropharynx	47 (68)
Hypopharynx	6 (9)
Larynx	5 (7)
CUP	6 (9)
Oral cavity	1 (1)
**T status**	
T0	6 (9)
T1	21 (30)
T2	23 (33)
T3	10 (15)
T4	9 (13)
**N status**	
N0	22 (32)
N1	12 (17)
N2a	4 (6)
N2b	26 (38)
N2c	5 (7)
N3	0
**M status**	
M1	1 (1)
**G status**	
G1	3 (4)
G2	33 (48)
G3	33 (48)
**R status**	
R0	49 (94)
R1	3 (6)
R2	0
**Therapy**	
Surgery	14 (20)
Surgery + RT	29 (42)
Surgery + C/RT	9 (13)
pC/RT	17 (24)
**Nicotine abuse**	41 (59)
**Alcohol consumption**	41 (59)
**Nicotine abuse and alcohol consumption**	31 (45)

## Data Availability

The data presented in this study are available on request from the corresponding author.

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
