# Peer review of "Point Shear Wave Elastography in Diagnosis and Follow-Up of Salivary Gland Affection after Head and Neck Cancer Treatment"

_jcm, 2022, doi:10.3390/jcm11216285_

Round 1

Reviewer 1 Report

Dear authors!

Overall, this is an interesting work.

Radiation-induced xerostomia is a really important conflict for patients.So any work dealing with this team seems useful.

Line 109: what characterizes an experienced medical examiner? Please explain in more detail.

Line 232: is there a cut off regarding the irradiated dose in Gy?

Author Response

Dear authors!

Overall, this is an interesting work.

Radiation-induced xerostomia is a really important conflict for patients.So any work dealing with this team seems useful.

Response: Thank you very much for this comment, we agree with the reviewer.

Line 109: what characterizes an experienced medical examiner? Please explain in more detail.

Response: We added more information into this section.

Line 232: is there a cut off regarding the irradiated dose in Gy?

Response: The cut off value was calculated based on the results of both groups, the patients and the healthy controls (and out of the 69 patients, who were included, 55 had received any kind of radiation, either primarily or additional, during their treatment). The stiffness of the gland, which is evaluated by this method, actually was smaller in patients who received lower dosages of radiation.

Reviewer 2 Report

The science behind this paper is interesting, though there exist several aspects of the study design that should be altered or improve to result.

I am entirely unfamiliar with point shear wave elastography and its medical applications; a more robust explanation of this technology and how it is currently being employed in the field of medicine would elucidate the reason this technology is being studied with regard to the salivary glands. Most importantly, if pSWE measures tissue stiffness, where is the data that increased stiffness of the salivary glands correlates with decreased salivary production? I know that this seems to be common sense, but unless there is data showing that increased tissue stiffness can be directly correlated with a formal measurement of decreased salivary flow, this seems like a large leap to make in asserting that pSWE can be used as a measurement of salivary function.  

With regard to the patient population, the oropharyngeal cancer population is really the population of most interest with regard to sparing the effects of radiation therapy, as many of these patients in the HPV population are being treated with surgery alone or de-escalated radiation therapy. Given that the vast majority of patients were oropharyngeal cancers, I would recommend limiting the data to this homogenous patient population and doing a case control study, matching the control group to align with the characteristics of the cancer patients. It was also not mentioned if the control group was screened for any history of salivary impairment or dysfunction (e.g. autoimmune disease) that may affect results.

The data is presented in a cohesive manner, but I do not fully understand the rationale for including the assessment of the liposomal agent in this study. I feel that this should be its own topic in its own paper once the validation of pSWE is confirmed; similarly, just because the glands are softer after the administration of the liposomal agent, how do we know that this infers improved salivary function without a formal measurement of salivary production/flow? 

Author Response

The science behind this paper is interesting, though there exist several aspects of the study design that should be altered or improve to result.

Response: Thank you very much for your evaluation and revision of our manuscript.

I am entirely unfamiliar with point shear wave elastography and its medical applications; a more robust explanation of this technology and how it is currently being employed in the field of medicine would elucidate the reason this technology is being studied with regard to the salivary glands. Most importantly, if pSWE measures tissue stiffness, where is the data that increased stiffness of the salivary glands correlates with decreased salivary production? I know that this seems to be common sense, but unless there is data showing that increased tissue stiffness can be directly correlated with a formal measurement of decreased salivary flow, this seems like a large leap to make in asserting that pSWE can be used as a measurement of salivary function.  

Response: The method of pSWE is in fact broadly used nowadays and especially part of the clinical routine in evaluating parenchymatous organs, mainly the liver. Here are some studies on this topic:

Piscaglia F, Salvatore V, Di Donato R, D’Onofrio M, Gualandi S, Gallotti A, Peri E, Borghi A, Conti F, Fattovich G, Sagrini E, Cucchetti A, Andreone P, Bolondi L (2011) Accuracy of VirtualTouch acoustic radiation force impulse (ARFI) imaging for the diagnosis of cirrhosis during liver ultrasonography. Ultraschall Med 32:167–175

Goertz RS, Sturm J, Pfeifer L, Wildner D, Wachter DL, Neurath MF, Strobel D (2013) ARFI cut-off values and significance of stan- dard deviation for liver fibrosis staging in patients with chronic liver disease. Ann Hepatol 12:935–941

There are much more current data available. However it is true, that these methods are currently not available in every country, especially in the U.S. sonoelastographical methods were not available for a long time and I am not entirely sure if they are yet. We added more information on this method into the manuscript. In addition, you can find studies on this using different search items, e.g. ARFI-imaning, shear wave velocity, virtual touch tissue quantification, depending on the company.

Regarding the correlation between SWE and salivary flow, you raise an absolutely correct and important point. So far, there have been studies that observe a similar development of salivary flow and SWE under local therapy (Hofauer, B.; Mansour, N.; Heiser, C.; Strassen, U.; Bas, M.; Knopf, A. Effect of liposomal local therapy on salivary glands in acoustic radiation force impulse imaging in Sjogren's syndrome. Clin Rheumatol 2016, 35, 2597-2601, doi:10.1007/s10067-016-3395-6) and higher SWE in patients with e.g. Sjögren's syndrome compared to a control group and salivary flow values that behave in the same way (Knopf, A.; Hofauer, B.; Thurmel, K.; Meier, R.; Stock, K.; Bas, M.; Manour, N. Diagnostic utility of Acoustic Radiation Force Impulse (ARFI) imaging in primary Sjoegren`s syndrome. Eur Radiol 2015, 25, 3027-3034, doi:10.1007/s00330-015-3705-4), but a direct correlation between these parameters has never been described and therefore we do not present this direct connection in our manuscript. We checked the manuscript again in order not to falsly claim this correlation.

With regard to the patient population, the oropharyngeal cancer population is really the population of most interest with regard to sparing the effects of radiation therapy, as many of these patients in the HPV population are being treated with surgery alone or de-escalated radiation therapy. Given that the vast majority of patients were oropharyngeal cancers, I would recommend limiting the data to this homogenous patient population and doing a case control study, matching the control group to align with the characteristics of the cancer patients. It was also not mentioned if the control group was screened for any history of salivary impairment or dysfunction (e.g. autoimmune disease) that may affect results.

Response: Thank you for this valuable comment. In fact, the sole inclusion of patients with an oropharyngeal carcinoma would make the group more homogeneous. We originally thougth that patients with carcinomas of other locations in the head and neck area are also affected by the same side effects and we therefore chose the inclusion criteria more generously. A retrospective exclusion would be difficult, but we add a note to the discussion and would consider this comment for further studies.

The short comment on the healthy controls in lines 136-138 has been extended.

The data is presented in a cohesive manner, but I do not fully understand the rationale for including the assessment of the liposomal agent in this study. I feel that this should be its own topic in its own paper once the validation of pSWE is confirmed; similarly, just because the glands are softer after the administration of the liposomal agent, how do we know that this infers improved salivary function without a formal measurement of salivary production/flow? 

Response: Thank you for this comment. After discussion with the co-authors before preparing the manuscript, we came to the conclusion that the inclusion of the results of the effect of the local therapy would be a useful addition, since it can be shown that the tissue condition measured using pSWE is subject to a certain modularity, which at least has no correlation with the factor time. Similar studies have also shown that local therapy can influence this objective parameter (Hofauer, B.; Mansour, N.; Heiser, C.; Strassen, U.; Bas, M.; Knopf, A. Effect of liposomal local therapy on salivary glands in acoustic radiation force impulse imaging in Sjogren's syndrome. Clin Rheumatol 2016, 35, 2597-2601, doi:10.1007/s10067-016-3395-6). In the study mentioned, also a trend towards a higher salivary flow rate was observed without reaching a level of statistical significance (p=0.059), which is a uniform development with the pSWE values (but in another population with another disease of course). We do not now if softer tissues automatically comes along with better salivary flow rates and we double checked the manuscript to avoid this conclusion – this should be part of future evaluations.

Additionally we modified our conclusion to better suit with the results as this was also mentioned by the reviewer within the revision process.